# Influence of Rheological and Morphological Characteristics of Polyhydroxybutyrate on Its Meltblown Process Behavior

**DOI:** 10.3390/ma16196525

**Published:** 2023-10-01

**Authors:** Tim Höhnemann, Ingo Windschiegl

**Affiliations:** German Institutes of Textile and Fiber Research (DITF), Koerschtalstr. 26, D-73770 Denkendorf, Germany; ingo.windschiegl@ditf.de

**Keywords:** sustainability, biopolymers, polyhydroxyalkanoates, polyhydroxybutyrate, polymer processing, fibers, meltblow(n), molecular characterization, rheology, polymer degradation

## Abstract

Polyhydroxybutyrate (PHB) is a promising biopolymer. However, processing PHB in pure form in thermoplastic processes is limited due to its rapid degradation, very low initial crystallization rate, strong post-crystallization, and its low final stretchability. In this article, we screened commercial PHBs for morphological characteristics, rheological properties, and “performance” in the meltblown process in order to reveal process-relevant properties and overcome the shortcoming of PHB in thermoplastic processes for fiber formation. An evaluation of degradation (extruded (meltblown) material vs. granules) was performed via rheological and SEC analysis. The study revealed large differences in the minimum melt temperature (175 up to 200 °C) and the grade-dependent limitation of accessible throughput on a 500 mm plant. The average fiber diameter could be lowered from around 10 μm to 2.4 μm in median, which are the finest reported values in the literature so far. It was found that the determination of the necessary process temperature can be predicted well from the complex shear viscosity. Different to expectations, it became apparent that a broader initial molar mass distribution (>8) is suitable to overcome the state-of-the art limitations of PHAs in order to stabilize fiber formation, increase the productivity, and obtain better resistance towards thermal degradation in process. Accordingly, longer polymer chain fractions could be more affected by degradation than medium and short polymer chains in the distribution. Further, a low initial narrow distributed molar mass resulted in too brittle fabrics.

## 1. Introduction

The use of bioplastics opens the potential for a fundamental contribution to future resource conservation. In this context, the term bioplastics is widely used imprecisely. Biodegradable polymers (which decompose under certain conditions only into CO_2_ and water as residues) are not necessarily also biobased (polymers on the basis of biomass or monomers derived from biomass) and vice versa [1]. A promising material class, combining both specifics, are polyhydroxyalkanoates (PHAs). They can be gained by biosynthesis in natural isolates, genetically engineered plants, different known genera of Gram-positive and Gram-negative recombinant bacteria [2], and various types of archaea [3]. These produce linear polyesters to store energy and carbon by intracellular inclusions [3,4]. PHAs are composed of R(-)-3-hydroxyalkanoic acid monomers with 3 to 14 carbon atoms and different molecular conformations (saturated or unsaturated and straight or branched chains with aliphatic or aromatic side groups), reaching 50,000 to 2,000,000 g·mol^−1^ [2,3]. More than 100 different monomer variations have been identified so far [3,5]. The PHA type, its molar mass, and yield depend on the microorganism “used” as well on the growth conditions and carbon source. The accumulated yield can reach up to 90% under nutrient (Mg, P, and N_2_) depletion (“stress”) [2,6].

PHAs produced by bacteria offer a sufficient molar mass to have similar characteristics to conventional plastics, e.g., polypropylene [6,7,8]. The best-characterized PHA is poly(3)-hydroxybutyrate (P(3)HB), a short-side-chain PHA (ssc-PHA). It delivers good properties, such as good use durability and UV resistance, and is water insoluble. A unique “feature” is the combination of biodegradability and hydrophobicity [9,10,11]. It can be hydrolyzed by many aerobic and anaerobic microorganisms in soil, compost, sewage sludge, freshwater, and even marine water [5], depending on the conditions, such as temperature, molar mass, and pH, as well as on its crystallinity [12,13]. Another benefit to other polymers is that the degradation product (3-hydroxybutyric acid) occurs in the metabolism of higher organisms [14] and is degraded without the release of toxic residues, unlike, for example, PLA [9,15,16].

Instead of gaining PHB by bacteria, which have to be fed with agricultural raw materials, its production by autotrophically growing cyanobacteria, also called “microalgae”, is particularly interesting. As in plants or algae, technically supplied or atmospheric CO_2_ is assimilated and actively converted into biomass by photosynthesis. Under suitable conditions, some of the fixed CO_2_ is converted to PHB and stored [17,18,19]. In contrast to polylactides or biobased polyolefins, this fulfills another key aspect of the bioeconomy: “neither additional agricultural area is required for this purpose, nor crude oil is consumed” [20]. Although PHB was discovered, isolated from bacteria, and characterized in the 1920s [21] and the transfer of production to an industrial scale started as early as the 1960s [7], with various commercial suppliers in the 1980s [22], the processing of pure PHB in thermoplastic processes is still challenging [22,23,24,25,26]. This is due to its rapid degradation, as the crystallite melt temperature is located close below the temperature at which thermal degradation starts [27]. Fiber formation from melt, in particular, is limited by additional factors [28], such as a high adhesion of the melt and a low melt strength and stretchability [29], as well as a very low initial crystallization rate combined with a strong post-(secondary) crystallization [30], resulting in a high crystallinity (60–80% [27]) with a large spherulite size [27,31]. This results in a further disadvantage for industrial use, the high brittleness [3,29]. For these reasons, pure PHB is special and is mostly processed via solution-based techniques or into three-dimensional injection-molded parts that still have only a very limited market. Of particular interest is the use of PHB for wound closure and healing [32,33,34,35]. Other examples of its use are in the development of cardiovascular products [36,37], drug delivery (microparticulate carriers) [33,38,39], nerve repair processes [32,37], soft tissue repair [32], and cell implants [3]. Furthermore, it is also used for agricultural applications [40], such as biodegradable plastic films for crop protection or seed encapsulation.

To achieve fibers successfully via meltspinning in a stable process, the use of organic or inorganic additives [15,41] or even graft copolymerization [27,42] as potential sources of nucleating agents, as well as circumstantial quenching methods [10,43] combined with complex drawing [27,42] and post-procedures [42,44,45,46], have been used to try to overcome the shortcomings of PHAs but have only been implemented on small scales in mono-hole spinnerets or for coarse fiber diameters [41]. Further, the production and use of PHA copolymers as superior alternatives to the “pure” PHAs are frequently reported. The most common is PHBV or, more accurately, [P(3HB-3HV)], resulting from the incorporation of 3-hydroxyvalerate (3HV) into P(3)HB, which can be achieved by co-feeding the substrate (valeric acid) to the producing bio-organism [2,47] (e.g., bacteria [46] and also plants [38]). It is stated to be less crystalline (55–70% [32]) and, in the following, less brittle but also less stiff [2,32,38,48]. Its properties can be varied according to the 3-hydroxyvalerate (3HV) content in the structure [36,40], and it is thus more flexible and easier to process, among other advantages, with a significantly lower melt temperature [9,42,49] (170 vs. 145 °C for 3% vs. 20% amount of 3HV [32]). Moreover, the blending of PHB, most often with PHV (poly (3-hydroxyvalerate), is pursued for the same purpose [2,22,33,50,51].

However, the use of copolymers cannot exclude all the problems of pure PHAs. Above all, the high costs are the main disadvantage compared to common polymers, even pure PHB, which has lower yields in synthesis [24,26]. The slow crystallization that also characterizes PHBV and its low degradation start temperature (170 °C) hinder the melt spinnability [9]. Another approach is melt blending of PHAs with polylactic acid (PLA), which allows processing on conventional machines, while it improves limits of PLA considering the final properties [52,53]. By bicomponent spinning of a PHBV core and a PLA sheath, Hufenus et al. successfully combined the advantages of both polymers to skip the disadvantages of the melts [23]. As a major disadvantage, resulting products were inseparable or barely separable material mixtures, which severely impairs biodegradability in particular.

Indeed, the conditions required for the production of fibers via meltblown are different to yarn spinning or spunbond processes and more “harsh” for the polymer concerning degradation [54] and they are not established for PHAs so far. For the classical meltblown process, the setting is based on the development of Exxon Mobile Corp. (Irving, TX, USA) in the 1960s [55]. The polymer melt stream flows out of a prismatic linear assembly of several hundred die orifices (“Exxon-type die”) and is taken up by two convergent streams of hot, primary process air with high velocity [56]. Along the distance to a collector (“DCD”), the air accelerates and stretches the fibers immediately. Together with the inflow of secondary air (around four times the amount of primary air [51,57]), a turbulent free jet is formed [58,59], in which the final fiber diameter of the fibers is reached [56]. The process requires a low molecular weight corresponding to a melt flow index (MFI) > 150 g·10 min^−1^ (melt spinning: 20–35 g·10 min^−1^) [55] as well as a narrow dispersity to achieve uniform fiber webs [56,60]. Although the industrial meltblown market is mainly focused on polypropylene and polyester, a high variety of polymers can be successfully processed. Recent trends clearly show that the demand for biopolymers for “green products” is high, but biodegradable polymers still pose a challenge due to their rheological behavior and low MFI. Therefore, publications on meltblown processing of biopolymers are still rare, with an existing focus on PLA [61,62,63,64,65,66]. A general study of meltblown processing of biodegradable polymers gives a publication of Müller et al. from 2001 [60]. Commercial types of PLA, polyesteramide (PEA), polyvinylalcohol (PVA), cellulosediaacetate (CDA), and polycaprolactone (PCL) were examined. Of these, only PLA and PEA proved suitable for the meltblown process, the latter, however, only with the addition of glycerin as a softener, as it could not be processed in pure form due to fiber shortage in the stream. For CDA, flow properties were reported to be insufficient, although they were well extrudable. Fiber formation was not possible such as for PVA and PCL. “Thermal overload” was found for all polymers by achieving excessive process temperatures in order to try to improve processability. Average fiber diameters above 10 μm could be achieved with PLA and PEA (10 µm with addition of 1% glycerin, 20 μm with 2% glycerin). However, “thermal branching” of the fibers was observed for both and the elongation of PLA webs was limited to a maximum of 2%. 

One study using PHB in the meltblown process was published by Sójka-Ledakowicz et al. in 2014 [25]. Nonwovens could be laid with fiber diameters in the web ranging from 14 to 45 μm but only in a small-scale set-up (polymer: 0.12 kg·h^−1^, air: 9 m^3^·h^−1^) far away from industrial process conditions (>10 kg·h^−1^ and several 100 m^3^·h^−1^). However, the obtained diameters of the nonwovens are in accordance with the range stated by Kann and Whitehouse (37 µm) [67]. Certainly, huge amounts of plasticizers (5–15%) were used in order to reach higher elongations of the web, which were 4% maximum. 

The presented advantages and disadvantages of PHAs, as well as the state of the literature on the requirements for biopolymers for successful meltblown processing on an industrial scale, leave a research gap for the aimed investigations of this work. In this study, we characterize various PHBs, which are commercially available on the market in reasonable amounts (several 100 kg) for their rheological and thermal behavior, record their molecular structure by size exclusion chromatography (SEC), and examine their processability and process behavior in the meltblown process using a technical-scale meltblown line. Nonwoven webs are characterized on their base weight, thickness, average fiber diameter, air permeability, and their mechanical performance (tensile test). Wide-angle X-ray-scattering (WAXS) and differential scanning calorimetry (DSC) are used to evaluate the crystallization behavior. MFI and shear rheological test on the extrudate are performed in order to reveal the degradation processes. Insights into the changes of the molar mass distribution and the rheological properties before and after processing will help to define the processability requirements of biopolymers and correlate material characteristics and their process windows with the performance of the obtained nonwovens.

## 2. Materials and Methods

### 2.1. Materials 

Three different commercial PHBs and one commercial PHBV were purchased and the following sample designations are used for the materials:

Mirell F1006 (PHB1), defined as injection molding grade with FDA approval, was acquired from Telles LLS (Orlando, FL, USA). It has a density of 1.3 g·cm^−3^ and a melt temperature of 160–165 °C [68].

P316 (PHB2) was obtained from Biomer Biopolyesters (Schwalbach, Germany) with a density of 1.20 g·cm^−3^ and an MFI (180 °C, 2.16 kg) of 10 (g·10 min^−1^) [69].

Emnat Y3000P (PHB3) and Emnat Y1000P (PHBV) from TianAn Biologic Materials Co., Ltd. (Ningbo, China) were purchased from Helian Polymers BV (Befeld, Netherlands). Both are designated as grades for extrusion and thermoforming, each with a density of 1.20 g·cm^−3^, a melt temperature of 175–180 °C, and an MFI (180 °C, 2.16 kg) of 10–25 g·10 min^−1^ [70,71].

### 2.2. Meltblow Set-Up

The meltblown system is a self-designed technical-scale line of 500 mm working width, consisting of a single-screw extruder (3 zone screw, ∅ 20 mm × 20 D) from Extrudex GmbH, Mühlacker, Germany) and a gear pump of Mahr Metering Systems GmbH (Göttingen, Germany) with a volume of 0.6 cm^3^·rpm^−1^ to melt and transport the polymer to the spinning beam with a maximum throughput of 4 kg·h^−1^. The air system consists of a compressor (Aertronic D12H) of Aerzener Maschinenfabrik GmbH (Aerzen, Germany) with an air volume flow limit of 220 Nm^3^·h^−1^ (minimal) and 325 Nm^3^·h^−1^ (maximal), combined with a flow heating system of Schniewindt GmbH & Co KG (Neuenrade, Germany). The spinneret is a 561-hole Exxon-type die with a width of 500 mm (28 holes per inch (hpi)) and nozzles of 0.3 mm in diameter (L/D = 8). The maximal die pressure of the spinneret is set at 50 bar with a safety limit of 45 bar. The set-back between nozzle tip and air blades is 1.2 mm and the end gap was set to 2.0 mm for all trials. The conveyor belt of Siebfabrik Arthur Maurer GmbH & Co KG (Mühlberg, Germany) is a steel fabric tape in canvas weave with clip seam in a total width of 0.72 m (No. 16·cm^−1^ linen weave) with a stainless steel (1.4404 AISI 316L) warp and weft wire of 0.22 mm in diameter. It has a maximal take-up velocity of 10 m·min^−1^ and can be adjusted in height relative to the die from 200 mm up to 500 mm to vary the die–collector distance (DCD). Below the belt section, where the filaments are laid down, an air-suction box (suction surface of 0.128 m^2^, 0.20 m × 0.64 m) with a maximal suction volume of 2900 Nm^3^ꞏh^−1^ (maximum flow velocity: 11 m·s^−1^) is placed to remove the process (and secondary) air. 

Variable parameters of the entire system (in the running process) are:Polymer throughput;Process temperature (melt);Process temperature (air);Air throughput;Die–collector distance (DCD);Collector speed.

### 2.3. Nonwoven Production

Meltblown trials were executed with the materials listed in Section 2.1 using the system described in Section 2.2, varying the process temperature and polymer throughput as the main parameters to reveal a stable process window. The die-to-collector distance (DCD) was varied between 200 and 500 mm in combination with an adjustment of the process air temperature to obtain an ideal deposition behavior of the fibers. This means that sticking or depletion of the deposit on the conveyor belt is avoided and the fiber-to-fiber bond is reduced. The collector speed was adjusted to the polymer throughput in accordance to produce a constant area base weight of the produced nonwovens of 100 gꞏm^−2^ in order to obtain comparability (without influence of the base weight) of web properties at different process settings. The process air throughput was varied between minimal and maximal output (220–325 Nm^3^·h^−1^) of the compressor in order to define the possible diameter range for each polymer at the respective process setting.

### 2.4. Determination of the Moisture Content

The residual water content for all polymers was determined by Karl Fischer titration, which was performed on an “899 Coulometer” and an “885 Compact Oven SC” (both: Deutsche METROHM GmbH & Co. KG, Filderstadt, Germany) at 140 °C. The resulting water content should be <150 ppm, respectively.

### 2.5. Polymer Characterization

Shear rheological experiments in time-sweep modes were performed on a “Physica MCR 501” rheometer (Anton Paar Group AG, Graz, Austria) in plate–plate geometry at different temperatures. Polymer granules were placed on the lower plate (25 mm in diameter) and the gap was adjusted to 1.0 mm. Afterwards, excess material was removed and the test was performed under nitrogen atmosphere (50 mL·min^−1^) with 10% strain and an angular frequency of 10 rad·s^−1^ for a period of 15 min, with the gap adjusted so that the normal force remained constant during the measurement.

Measurements of the melt flow index were performed on the samples at 190 °C with a load of 2.16 kg according to ISO 1133 using a “Göttfert MI-B” (GÖTTFERT Werkstoff-Prüfmaschinen GmbH, Buchen, Germany). Depending on the flowability (~experiment time) 10–15 data points were taken with constant time steps of seconds and the mean value calculated. Three measurements were performed per sample.

For size exclusion chromatography (SEC) an Agilent Technologies 1260 Infinity II High Temperature GPC System (GPC 220, Agilent Technologies, Inc., Santa Clara, CA, USA) equipped with a refractive index detector was used and operated at 50 °C in m-cresol as eluent. The PHA materials were dissolved in an m-cresol (250 mg·L^−1^ Ionol) solution at 80 for 0.5–3 h. Three consecutive PLgel Olexis columns (0.013 Å pore size) and one precolumn were used while applying a flow rate of 0.4 mL·min^−1^. The measurement time was 100 min per sample with an injection volume of 100 μL and a flow rate of 1 mL·min^−1^. For the recording and evaluation of the chromatograms, the GPC/SEC (size exclusion chromatography) software of Agilent Technologies (Santa Clara, CA, USA) was used. Narrow distributed polystyrene standards with molar masses from 1681 to 2,000,000 g·mol^−1^ were used for calibration and the evaluation was made against the 3rd order using Mark–Houwink parameter “κ” of 14.1·10^−5^ dl·g^−1^ and a Mark–Houwink exponent “α” of 0.7 without flow rate correction.

Thermal properties were characterized via differential scanning calorimetry (DSC) and thermogravimetric analysis (TGA). DSC measurements were carried out under nitrogen (50 mL·min^−1^) on a Q2000 differential scanning calorimeter (TA Instruments Inc., New Castle, DE, USA). The sample mass was around 2 mg. The melt enthalpy ΔH_m_ and melting peak temperature T_m,p_ were determined from the heat flow–temperature curves of the first heating cycle, as well as the glass transition temperature T_g_. The recrystallization enthalpy *ΔH*_R_ and recrystallization peak temperature T_R,p_ were determined from the first cooling cycle. One measurement per sample was carried out performing two heating–cooling cycles from room temperature to 200 °C, applying a heating and cooling rate of 10 K·min^−1^. The degree of crystallinity *χ_c_* was calculated by standardizing the melt enthalpy to the standard melt enthalpy *ΔH_m_*_,0_, as shown by Equation (1).
(1)$$\chi_{c}=\frac{\Delta H_{m}}{\Delta H_{m,0}}$$

The standard melt enthalpy used was 146 J·g^−1^ according to the literature [5,72].

TGA measurements from room temperature to 600 °C with a heating rate of 10 K·min^−1^ under nitrogen atmosphere (60 mL·min^−1^) were performed using a “TGA Q5000” of TA Instruments (New Castle, DE, USA) to obtain information on the onset and kinetics of thermal degradation. The mass loss was recorded and analyzed with Universal V4.5A of TA Instruments (New Castle, DE, USA). 

### 2.6. Nonwoven Testing

The area base weight of nonwovens was determined by cutting out and weighing square sections of 10 × 10 cm (100 cm^2^) out of the nonwovens according to DIN EN ISO 29073-1. To analyze the homogeneity of the nonwoven, three samples were taken across cross direction (CD). In accordance with the base-weight sampling, the air permeability was measured on the 10 × 10 cm sections in accordance with DIN EN ISO 29073-3, with a sample size of 20 cm^2^ and a differential pressure of 200 Pa. On the same samples, the nonwoven thickness was measured according to DIN EN ISO 9073-2 using a test head (Frank-PTI GmbH, Birkenau, Germany) of 25 cm^2^ and a test force of 5 cN·cm^−2^. Eight measurements were executed diagonally along the sample.

Tensile tests were performed on an Instron “4301” tensile tester (Instron GmbH, Darmstadt, Germany) to determine the tensile strength (σ_m_), the elongation at peak force (ε_m_) of the fabrics in machine direction (MD) and CD, respectively, as well as the elastic modulus (E) as secant modulus. For each sample, 5 specimens with a width of 15 mm were cut out in MD and CD. Their thickness was determined individually according to DIN EN ISO 9073-2 and the median of five measurements was used for the calculation of the stress from the recorded force. The tests were executed with 100 mm·min^−1^ using a 5 kN measuring head with pneumatic clamps (100 mm clamping length).

The fiber diameter distribution was determined using scanning electron microscopy (SEM). Therefore, a round sample was punched out of the nonwoven and placed on the SEM carrier, which was sputtered with a gold–palladium layer of 10–15 nm. Three SEM micrographs per sample were taken with a magnification of ×1000 using a “TM-1000 tabletop electron microscope” of Hitachi High-Tech Corporation (Tokyo, Japan). The magnification was chosen to catch around 40 single fibers and contrast and illumination were adjusted to gain an image of straight monochromic fibers against a dark monochrome background. To analyze the images, the beta-software “MAVIfiber2d” of Fraunhofer ITWM (Kaiserslautern, Germany) was used [73]. First, the images were smoothed by an algorithm and binarized by the software before a statistical analysis was performed over each fiber pixel without segmentation into individual fibers [74,75]. After merging the output of the three images, the mean and median fiber diameter as well as the standard deviation and interquartile range were given out.

XRD measurements were recorded on a D/Max Rapid II diffractometer (Rigaku Corp, Akishima, Japan) using monochromatic Cu Kα radiation (*λ* = 0.15406 nm; U_acc_ = 40 V; I_acc_ = 30 mA) and an image plate detector. A scanning rate of 0.2°·min^−1^ and a step size of 0.1° were used. The measurement time was 1 h for all investigated samples. The obtained scatter images were converted to the corresponding diffractograms using 2 theta(*2θ*)-intensity conversion by the software 2DP (Rigaku Corp, Akishima, Japan). The diffraction patterns were analyzed using the PDXL 2.0 software, and pseudo-Voigt profile fitting was chosen for the evaluation of reflex positions and crystalline fraction determination. The degree of crystallinity *χ_c_* was calculated according to Equation (2):(2)$$\chi_{c}=\frac{\sum I_{c}}{\sum \left(I_{c}+I_{a}\right)}$$
where I_c_ and I_a_ are the integrated intensities of crystalline reflexes and amorphous reflexes, respectively.

The samples were prepared as follows: nonwoven fabrics were pressed and arranged parallel on the carrier.

### 2.7. Characterization of Degradation

The degradation of the polymer material during the meltblown process was characterized via three different characteristics. Nonwoven material was crushed and remolten in the plate–plate rheometer for performing time sweeps of 3 min duration (according to the procedure in Section 2.5). The complex viscosity data point after 30 seconds was taken as reference for the comparison in order to estimate the degradation. In the same way, the material was prepared for MFI measurements (ISO 1133, at 190 °C, 2.16 kg). Further nonwoven material was dissolved an m-cresol to determine the molecular mass distribution by SEC (as described in Section 2.5, respectively).

## 3. Results 

### 3.1. Raw Material Characterization

To pre-estimate the processing window and to reveal possible differences in processing behavior, the materials were extensively investigated for their thermal behavior, macroscopical properties, and rheological characteristics. 

#### 3.1.1. Thermal Properties

The decomposition behavior of the materials was analyzed via mass loss using a TGA temperature ramp, as shown in Figure 1.

All four samples show the highest mass loss rate in the range of 300 °C (±10 K). PHB3 and PHBV, which were obtained from the same supplier and have similar data sheet specifications [70,71], show a later onset of decomposition than PHB1 and PHB2, followed by a severe mass loss of almost 100% until 310 °C. Here, the PHBV shows a slight shift of the whole curve to lower temperatures. PHB1 has an earlier onset at about 200 °C, with a lower decomposition rate up to about 280 °C (~10% mass loss), followed by complete decomposition up to 300 °C. PHB2 shows the broadest decomposition range of the four samples with an even earlier decomposition onset, already around the nominal melting temperature of 170 °C, but with only moderate mass loss up to 290 °C (50% mass remaining) and still remaining decomposition (10%) between 315 and 450 °C.

In the literature (e.g., [53,76,77]), the thermal degradation of PHB is referred to as almost exclusively a nonradical random chain scission reaction (cis-elimination) involving a six-membered ring transition state [76,77]. This results in a rapid decrease in its molecular weight [53]. Further, the thermal stability was found to depend on the size of the counterion at the PHB endgroup, which has the form of carboxylic acid salts with Na^+^, K^+^, and Bu_4_N^+^ counterions. Based on that, the degradation via intermolecular α-deprotonation through carboxylate is assumed to be the main PHB decomposition pathway at moderate temperatures (above 120 °C) [78].

Synthesis of PHBV is one possible way to overcome this temperature sensitivity [3]. Hydroxyvalerate (HY) units in the PHB-polymer backbone are reported to lower the melting point and, thus, the processing temperature [79]. It is also expected that PHBV is less affine to thermal degradation [80,81,82] However, the degradation procedure proposed for PHB was supposed to be equally valid for PHBV with low HV contents [79]. 

The PHBV sample in this study shows no lower melting point compared to the PHB of the same supplier. Further, the degradation in TGA proceeds almost equally with increased temperature. However, the decomposition takes place even slightly faster. Additionally, of note is that the TGA curve of PHBV is a one-step curve like that of PHB. This is a further indicator that PHBV contains only a marginal HV content (compare [81]). 

Further, differences in the melting and (re-)crystallization behavior were examined by DSC. Figure 2a shows the significant range (as the glass transition of PHB is located at around 0 °C, it was not recorded by our measurements starting from room temperature) of the first heating cycle with the melting peak.

Both the positions and the expression of the four melt peaks are again different. While the peak of PHB3 and PHBV is again almost congruent (slight shift to lower peak temperature of PHBV), the peak of PHB1 occurs shifted to a lower temperature and shows a lower melting enthalpy. The melt peak of PHB2 is also shifted to a lower temperature but again with a broader, less sharp form of the peak and an earlier onset of the crystal melting at about 150 °C. This can be correlated to a broader molar mass distribution. 

The first and the second heating/cooling cycle are shown exemplarily for PHB2 in Figure 2b. The sample shows a recrystallization peak in the first cooling cycle. The melting peak in the second heating cycle is shifted to a lower temperature and also changes into a double-peak structure. Moreover, the recrystallization peak is shifted significantly to a lower temperature from approx. 110 °C to approx. 50 °C and thus from a sharp to a broader peak area. This behavior is representative for all materials used and can be attributed to the low degradation start temperature (170 °C) of PHB, such that the melt remains above the decomposition temperature for almost an hour (per run) during the heating and cooling to/from 200 °C under the test conditions with 10 K·min·^−1^. As revealed by the TGA curves, the polymers show a different tendency to decompose. Following from this, the data after the first heating cycle (first cooling cycle as well as the second heat/cooling run) were excluded from further analyzation as they are affected by decomposition of the polymer, which was present for all four materials.

The evaluation of the DSC heating ramp and characteristic indicates the TGA mass loss, which is given in Appendix A.

#### 3.1.2. Molecular Characteristics

To relate the differences obtained by thermal characterization to the polymer structure, the molar mass distribution was recorded by SEC, which is shown in Figure 3. 

As indicated by the DSC and TGA curves, the peak of PHB1 is shifted to a lower molar mass compared to PHB3 and PHBV, which are congruent. However, the dispersity is only slightly higher. Furthermore, PHB2 shows a very broad molar mass distribution with a dispersity of >8, containing the lower molar mass fractions of PHB1 and higher molar mass fractions than PHB3. It is also worth mentioning that the sample with the significant lowest molar mass (PHB1) also shows the lowest degree of crystallinity (obtained by DSC; Table A1). It can be concluded that (a) the majority of polymer chains (<10.5 g·mol^−1^) are too short to contribute significantly to crystal formation or (b) very short fractions present in the distribution (<10^4^) even hinder crystallization by acting as a “spacer” between longer chains.

#### 3.1.3. Rheological Behavior

Due to the decomposition behavior of PHAs already just above the melting temperature, the complex viscosity measurements from amplitude sweeps, frequency sweeps, or temperature sweeps could not be used as they were affected by the decline in the viscosity and elasticity/loss moduli over measurement time. 

Due to that, time sweeps at constant shear rate and strain were executed at different temperatures. For comparison of initial properties of the different samples, only the time-unaffected start values were taken. The plots of the absolute value of the complex shear viscosity are given in Figure 4.

According to the differences in the molar mass (see Section 3.1.2), the flow behavior of the different PHAs is also quite different. PHB1 (Figure 5a) has the lowest number and weight average of molar mass and shows the lowest viscosity, right above the melting temperature (see Table 1), but also has a quite stable viscosity over time compared to PHB2 (Figure 4b). PHB1 is already located within the desired viscosity range for the meltblown process [54], which is set by typical residence times in the extruder between 5 and 10 min and an upper viscosity limit between 100 and 150 Pa·s^−1^. PHB2 with a higher initial viscosity also “enters” this range, albeit only after about 500 s, showing a faster degradation as indicated before by the TGA curve by the earlier onset of decomposition (Section 3.1.1). At 180 °C and 185 °C, the viscosity is within the process window. PHB3 and PHBV show a high viscosity at 175 °C and 180 °C (PHBV not yet completely molten at 175 °C), as can be seen in the comparison in Table 1.

The MFI at 190 °C was measured to obtain an impression for the range of this more industrial-used value of the four polymers, which is quite equal for PHB1 and PHB2 as well as for PHB3 and PHBV. More differences can be obtained by the shear-rheological characterization, such as an estimation of the process window as indicated in Figure 4c,d. The estimated process temperature Tproc as well as the initial absolute value of the complex viscosity η_to_ after 300 s measurement time (η_300s_) as the characteristic residence time for the meltblown process (extrusion step) are shown in Table 2, together with the initial storage modulus (G′_t0_) and loss modulus (G″_t0_).

According to this shear-rheological characterization, PHB1 can already be processed just above the melting temperature at 175 °C, where degradation is still low. PHB2 shows a suitable flow behavior at 180 °C and 185 °C, while the estimated process temperature for PHB3 is already 190 °C. For PHBV, the viscosity at 195/200 °C would fit to the process range but also shows the fastest viscosity decline (>50% over 120 s measurement time). This would be contrary to the benefits for processing of PHBV mentioned in the literature [9,32,42,49].

### 3.2. Examination of Meltblown Processability

Based on the rheological characterization, the process temperatures of the melt were adjusted by the spinneret temperature to obtain the required flowability. Therefore, the die temperature (Heat Zones 6–8) was set around 5 K higher than the melt temperature, respectively. A heating ramp was applied from the feed zone (Zone 1: T = 160 °C) along the three extruder zones (Zone 2: T= 175 °C to Zone 4: T = Zone 6) to melt up and homogenize the materials and keep the thermal decomposition as low as possible. The process settings of the trials are summarized in Table 3.

#### 3.2.1. PHB1

At the estimated minimal process temperature, PHB1 showed a good processability concerning a continuous and homogeneous melt stream out of the capillaries, stable fiber formation/take-up by the air stream, and a homogeneous laydown on the conveyor belt. The throughput could be varied without limitations up to the maximal productivity of the set-up (3.5 kg·h^−1^ ≜ 0.11 g·ho^−1^·min^−1^) as the die pressure resulted at a moderate and constant level. However, the DCD had to be adjusted to the maximum of 500 mm because the fibers on the belt below the die were very sticky and glued to the conveyor belt, especially at the maximum throughput of 0.11 g·ho^−1^·min^−1^. This could be reduced by lowering the throughput. 

It can be seen from the SEM images (Figure 5) that the deposited fibers remain quite coarse and irregular in diameter, especially at a low process temperature (Figure 5a,b). With increasing throughput, the fibers start to flow/glue together (see Figure 5a). By raising the process temperature, the deposited fibers resulted in a more uniform diameter (Figure 5c,d). However, it was not possible to use a higher volume flow of process air as the fibers immediately glued irreversibly on the conveyor. 

Despite the good processability in terms of initial fiber formation and deposition, the nonwoven material of all PHB1 trials cured along the path from deposition to rewind. In the following, the fabrics were not useable/unrollable due to a high apparent brittleness. The embrittlement increased over time after winding (overnight). By touching the material, it disintegrated immediately into crumbs, so that no further characterization (air permeability, base weight, homogeneity, thickness, and thermal characterization) could be applied. This post-crystallization behavior is well known for PHAs, especially in yarn spinning processes. However, small rounds for the SEM examination could be punched out with great care.

#### 3.2.2. PHB2

PHB2 could be processed with a higher variability of the process settings. The fiber formation and “air-take-up” at the die was again very homogeneous, as was fiber deposition at all settings. In contrast to PHB1, the fabrics did not show curing induced by post-crystallization and were manageable after winding (also after days). Although the maximal applicable productivity was 0.077 g·ho^−1^·min^−1^ (2.6 kg·h^−1^), it was used as reference throughput according to the trials of PHB1 and due to the fact that very coarse fibers were formed at the lowest process temperature. In the following, the process temperature was raised under additional variation of the throughput, air temperature, air volume flow, and DCD. 

Starting with a melt temperature of 175 °C, a stable and homogeneous process was present at the spinneret. Altogether, the nonwovens showed high brittleness and a discontinuous fiber formation during deposition. The brittleness can be indicated by the breakage of individual fibers, representatively shown at 500-times magnification in SEM for PHB2-02 in Figure 6a.

Significantly, the expression of brittleness was lower than in all PHB1 samples and no (significant) post-crystallization was evident in the process. The fabrics also remained manageable even after days of storage. However, the deposited fabrics showed a very rough surface with an open structure and coarse fibers (trial PHB2-01 and -02). This can be seen in the SEM images at 100- and 500-times magnification in Figure 6b,c for maximal and minimal air volume flow (set by compressor power). The reduction in the throughput (PHB2-03 and -04) leads to a visually denser structure with smoother haptics. As shown in Figure 6d,e, this can be referred to as the formation of finer fibers as the air-to-polymer ratio is doubled and the force applied by the air acts more strongly on the individual fibers. For both settings, the resulting fibers were quite uniform in diameter and started to coalesce or merge into a branched structure at a higher air volume. Further, the fibers start to deliquesce and form branched drops (Figure 6e). The same was already reported as “thermal branching” in the literature for PLA and PEA meltblown fabrics [60]. With the higher air amount, the fabric also shows gluing/adhesion to the conveyor belt at both edges of the deposit width.

The increase in the process temperature as well as the air temperature by 5 K (180/185 °C; PHB2-05) resulted in less brittleness of the fabrics and better haptics and handling. Certainly, the merging of the fibers intensified with the higher melt temperature (see exemplary in Figure 6e), so that the DCD had to be raised to the maximum of 500 mm (and kept over all further trials). 

As the merging of fibers was still present and formation of shots also occurred, the temperature of the process air was reduced below the temperature of the melt (PHB2-07). Thus, the shot formation was eliminated. However, the fabric’s structure was “more open” and stiff due to the increase in the fiber diameter (see Figure 6g,h). Indeed, increasing the air volume flow and reducing the polymer throughput (0.039 g·ho^−1^·min^−1^) led to the best-quality fabric deposition so far, concerning a consistent dense nonwoven structure, soft haptics, and a more or less uniform fiber diameter distribution without merging of fibers (see Figure 6i,j).

A further stepwise (5K) decrease in the air temperature showed counterproductive effects. At an air temperature of 170 °C (PHB2-09), the fiber diameters became more irregular again and, at 160 °C (PHB2-10), cracks could be observed on the fiber surfaces (see Figure 6j). The use of a higher air volume did not contribute to the processability, as shot formation occurred. 

Critical shot formation (regular deposition from various die positions) was achieved by further increasing the melt temperature to 185 °C (PHB2-11, Figure 6k,l).

#### 3.2.3. PHB3

Compared to PHB1 and PHB2, PHB3 showed difficulties in the meltblown process. At the estimated process temperature range, the process pressure (at die) was significantly higher compared to the previous PHB types due to the higher molar mass distribution characteristics (number and weight average). At 185 °C (melt) and throughputs of more than 0.045 g·ho^−1^ min^−1^ (1.3 kg·h^−1^), the die pressure exceeded the critical value of 50 bar defined for the integrity of the spinneret. By raising the temperature to 190 °C, this critical pressure resulted above 0.60 g·ho^−1^ min^−1^ (1.9 kg·h^−1^). At flow rates of less than 0.04 g·ho^−1^ min^−1^, the initial die pressure declined very fast (several bar per minute) due to thermal degradation of the material. Moreover, shots were present at this temperature.

In the following, a constant die pressure could only be obtained for single process points, which can be considered as the equilibrium point between the degradation and the critical viscosity. However, all fabrics showed a significant yellow coloring due to the present degradation compared to PHB2. Additionally, only the maximum air volume flow could be used to depose a considerable nonwoven structure as shots and die adhesions occurred at lower air volume flow values. Further increasing the melt temperature was not an option as the decomposition and the formation of shots increased. On the other hand, at the same process temperature as PHB2 (180 °C of melt), no nonwoven structure was obtained, since only filaments that were too solid to be taken up by the process air were collected on the conveyor belt. 

The SEM images of these respective trials are given in Figure 7.

Compared to PHB1 (Figure 5) and PHB2 (Figure 6), the SEM images of PHB3 are more similar to those of some polyolefin meltblown fabrics (e.g., polypropylene). The formation of fine fibers can be observed. However, these form only a part of the fiber distribution, since the inhomogeneous stretching of the formed fibers by the process air results in a mixture of coarse and fine fibers. This can be reasoned through the higher initial viscosity (Figure 4) and the fact that decomposition does not take place uniformly. Therefore, it has to be noted that no flowing/merging of the fibers was observed for PHB3.

#### 3.2.4. PHBV

The processing trials of PHBV are similar to the meltblown process of PHB3 but, as indicated by the rheological characterization, with a shift to a higher process temperature. Starting from the same process temperature of PHB3 at 190 °C, only filaments that were too solid for air take-up could be extruded. At a melt temperature of 195 °C and 200 °C (with 5 K lower air temperature), only one setting for the throughput was found (see Table 3) that delivered an uncritical die pressure, which stayed stable without (fast) decline by thermal decomposition. 

The collected nonwoven samples were more handleable, had higher flexibility, and a softer haptic. However, reducing the amount of air immediately caused adhesion of melt at the die. The yellowing of the fabrics was also reduced but present compared to PHB3 (compared to PHB2). The SEM images (Figure 8) again showed fibers with significantly smaller diameter compared to PHB1 and PHB2. 

Again, a proportion of significantly coarser fibers or rather nonuniform stretching by air has to be noted. All in all, contrary to expectations, PHBV showed neither benefits in the meltblown process nor better fabric properties than PHB.

### 3.3. Nonwoven Characterization

The characteristics of the produced nonwoven fabrics of PHB2, PHB3, and PHBV are summarized in Table 4. For PHB1, only fiber diameter could be characterized, as described already.

The base weight of all samples was in acceptable agreement with the target range of 100 g·m^2^. Deviations occurred mainly in PHB2 with lower throughput (air force can act more effectively, leading to wider deposition) and in PHBV (narrower deposition width due to less flowable melt). The homogeneity of base weight along CD, as measured by the coefficient of variation (CV) of the fabrics, was also in an acceptable range between 8% and 22%.

High variations were obtained for the average fiber diameter as well as for the fiber diameter distribution (ratio of median and mean value). However, the achieved diameter averages are below the ranges, which were obtained via meltblown of PHAs in the literature so far [25,67]. Additionally, achieving fiber diameters below 3 µm in average enters the realm of several fossil-based polymers and opens a potential for PHB meltblown fabrics to be competitive in applications and move beyond scientific status. This is underlined by the fact that no further additives such as plasticizers were used in our work so far and the experimental scale has already left lab-scale.

In principle, the fiber diameter changed as expected with the adjustments of process properties, such as a decrease in diameter with increased air flow (e.g., PHB2-01 vs. PHB2-02, PHB2-03 PHB2-04, PHB2-06 vs. PHB2-07 and PHB3-01 vs. PHB3-03) and, conversely, an increase with decreased process air temperature (e.g., PHB2-05–PHB2-06–PHB2-09–PHB2-10). As the melt temperature increased, the diameter changed only slightly or not at all, because the fiber deliquescence started and was stronger on the conveyor belt with decreasing viscosity, which caused an opposite effect.

The lowest median and mean fiber diameters were obtained for PHB3 at 2.6 and 5.0 µm, respectively. As in PHB3, the lowest fiber diameters (4.6 µm in median and 7.0 µm in mean) were also obtained in PHB2 when the air temperature was set 5 K lower than the melting temperature, which seemed to be the most effective condition for the air to “act”.

These two samples also showed the best mechanical properties in both MD and CD. In general, it is also noticeable that PHBV formed fabrics with higher thickness (with only marginally higher base weight) and worse mechanical properties.

However, the MD tenacity values varied between 0.5 and 1.1 N·mm^−2^ as well as the elongation between 1% and 3%. This agrees with the review of Müller et al. for meltblown PLA fabrics [60]. Although Kann and Whitehouse exceeded this range by using huge amounts of plasticizer [67], they actually were limited too by a maximal elongation of 4%. This value was also reached in this study in PHB2 (sample PHB2-08). Actually, the elongation of the PHB2 trial PHB2-08 showed extended stretchability, combined with the highest MD and CD tenacity (1.6 and 0.9 N·mm^−2^).

A higher comparative significance can be obtained by the elastic modulus, which varies between 28 and 96 N·mm^−2^ for all collected samples and correlates mainly (but not only) with the fiber diameter. Since elasticity also correlates with the haptics and handling of the nonwovens, it can be used as a robust parameter comparing the fabrics to each other and to other nonwoven media. 

The air permeabilities also cover a broad spectrum from 680 to >7000 L·m^−2^·h^−1^ and are thus also in the range of meltblown media of standard polymers (~200 to ~2000 L·m^−2^·h^−1^). Its major influence factor is the fiber diameter, as coarse fibers (>10 µm) also formed very open nonwoven structures. However, the merging of fibers on the conveyor must also be taken into account, as it affects the nonwoven density.

In addition to the standard characterizations for nonwovens (base weight, thickness, and air permeability) and the mechanical testing, three exemplary samples of PHB2 and PHB3 were analyzed via WAXS. An exemplary X-ray diffraction image and the diffraction patterns of the tested samples are given in Figure 9a,b.

The X-ray diffraction images of all three samples show a discrete crystalline signal, given in Figure 9a for PHB02-10. PHB2-08 was produced with the half throughput compared to PHB2-10 and, thus, had less mass to cool in the air stream/to crystallize. However, both signals do not differ from each other and also not from the sample of PHB3, produced at higher temperature. This can be also seen in the diffraction pattern in Figure 9b, which shows the same crystalline structure with identical peaks. 

The diffractograms of the PHB nonwovens were evaluated by fitting (pseudo-Voigt profiles) in the range of 2*θ* = 10–40° In this way, the areas of the reflections could be determined and the crystallinity semi-quantitatively fixed at 75 to 80%. This corresponds to the impression that, unlike PHB1, no further post-crystallization could be observed in these three materials. Unfortunately, the latter could not be measured due to the high brittleness of the samples. However, the results indicate that the conditions in the meltblown process seem to be sufficient for a complete crystallization until the deposition, at least for the three materials that have a certain molar mass or, more precisely, high molar mass fractions (>10^6^ g·mol^−1^) in their molar mass distribution. This was not the case for PHB1, which showed post-crystallization after the deposition. This behavior is well known from the melt spinning of PHA fibers [30]. 

From the respective positions of the diffraction peaks of PHB2, PHB3, and the PHBV, it is evident that mainly the α-form is present. A comparison with the diffractograms of PHB films in the literature [5] shows that the shoulder at 2*θ* = 20° may indicate the presence of β-form crystals. However, no clear statement can be made on the basis of the accessible diffractograms but results agreed with the literature on electrospinning of PHB [5]. Here, the α-form was also determined as presenting crystal structure, agreeing with further literature on crystal formation of PHB from melt or solution crystallization [72]. Electrospun nanofibers (100–400 nm in diameter) showed a degree of crystallinity of 55%, with an additional peak indicating the presence of β-form crystals in WAXS-spectrum. This WAXS peak is typical for introduced orientations, such as uniaxially stretching [5,72], or is based on different possibilities: (a) distribution of different crystal sizes among others by random chain scission during degradation [77,83] and, thus, crystals with different lamellar thickness and thus different melting kinetics and (b) melting of different crystal structures (e.g., lamellas and spherulites), as already known for PA6 [84,85,86] and modified polyesters [87,88,89,90]. This is consistent with published findings on melt spinning showing orthorhombic (α) and hexagonal (β) crystals, the latter formed by stress-induced crystallization [13,46].

### 3.4. Evaluation of Degradation in the Meltblown Process

The degradation of PHB2, PHB3, and the PHBV was quantified via measurement of the molar mass distribution of the nonwoven material and by quantifying the flow behavior via determination of the MFI, as shown in Table 5.

The weight averages of the molar mass distribution show that a significant degradation took place during the metlblown process for all three materials. However, the number average of the molar mass of PHB2 almost stayed constant, while it was almost halved for PHB3 and PHBV. This is well illustrated in the following plot in Appendix B.

In addition, the dispersity of PHB2 was more reduced (8.5 to 6.1) than the other two materials (~4.5 to 3.5). This is also emphasized by the comparison of the molar mass distribution curves in Figure 10.

A very interesting insight is that the overall molar mass distribution of the initially more narrow distributed materials PHB3 (Figure 10b) and PHBV (Figure 10c) shifts more or less to lower molar masses. Whereas the change in the molar mass distribution of PHB2 (Figure 10a) is characterized by an increase in the fraction of intermediate molar mass, the number of fractions with low molar mass seems to stay constant and does not shift. The highest molar mass fractions disappear and seem to be mainly affected by process degradation. The superposition of the curves (Figure 10d) shows that the curve peak of PHB3 and PHBV, which was initially at a higher molar mass than that of PHB2 (compare Figure 3), is at a lower molar mass after processing. This may be correlated to the more stable process behavior for PHB2. A similar effect was shown by Schmack et al. [10] using high molar mass type with a molar mass (viscosity-average *M_v_*) of 360,000 g·mol^−1^ for the melt spinning of pure PHB. The spinnability and high drawability (draw ratio up to 6.9) were preserved, but the processing step decreased the intrinsic viscosity (i.V.) from 2.5 to below 1.5 dl·g^−1^ as well as the *M_v_* to 175,000 g·mol^−1^. Further electrospinning of PHB was executed by Mottin et al. [5], also using very high molar masses (*M_w_* of 600,000 g·mol^−1^). 

In order to obtain an impression/estimation for the reusability (recycling) of the PHB nonwoven material for reprocessing as rPHB or in a PHB:rPHB blend, the initial complex shear viscosity of the raw material (molten granule) is compared to the processed material (remolten nonwoven) in Figure 11.

As expected from the SEC data, the complex shear viscosity of the nonwoven samples dropped significantly. For PHB2, it can be seen that, at a melting temperature of 175 °C, the degradation was lower. However, as the processability was worse, it was not analyzed further by SEC, nor was the sample at 185 °C, which indicated less degradation at higher throughputs. Although the zero-shear viscosity is still in the useable range for the meltblown process, further irreversible degradation in a second life cycle will be too rapid for a use as 100% “rPHB”. The same will be the case for the PHB3 nonwoven material considering the raw materials’ viscosity drop over time, as shown in Figure 5.

However, the fact that the lower mass fractions of PHB2 were not affected/shifted to lower mass, it may be an approach to use it in a compound and broaden the molar mass distribution of a virgin PHB grade. 

Summarized, the results indicate that broad dispersity is beneficial for the meltlbown process of PHAs, contrary to common process understanding [73,83], e.g., for polyolefins or polyesters, which require rather narrow dispersity. It seems that (very) high molar mass fractions are more affected by the chain scission, which takes place randomly during the degradation process. They seem to “protect” the scission of lower molar mass fractions, which are initially required to obtain the required flowability at a process temperature as close as possible to the crystallite melting temperature, while the intermediate molar mass fraction retains the process stability. This was not achievable with a narrow low molar mass distribution, which seemed to cool down too fast and showed unmanageable post-crystallization. However, it should be noted that the lowest average fiber diameter was obtained with the narrow high molar mass distribution. In order to achieve a uniform distribution of finest fiber diameters, the broad distribution could be counter productive or an equally broad distribution shifted toward the lower molar mass could be more beneficial.

## 4. Discussion and Conclusions

Successful deposition of nonwovens from PHB by meltblown beyond a laboratory scale and without using additives was demonstrated. Experimental meltblown trials using different industrial PHAs, combined with their characterization by shear-rheological time sweeps and SEC, revealed the requirements for a successful and stable processing on both the material and process side.

Fiber diameters below 10 μm were achieved. Median diameters resulted in the range down to 2.4 μm, which is outstanding compared to the literature (>14 μm). Moreover, the median diameter could be varied up to 20 μm, enabling an interesting variation in width for different applications. 

Since the variance in the tenacity and maximum peak elongation within the processed fabrics was quite low, the elastic modulus proved to be a comparable and reliable parameter for the mechanical properties of PHA fabrics, correlating with the fabrics’ haptics and further handling.

It was demonstrated that the basic processability in terms of extrusion and fiber formation at the spinneret can be estimated by the complex shear viscosity by measuring the static (at constant shear and strain) absolute value of complex viscosity over time. It was found that the processability is better when a broad molar mass distribution (>8) with (very) high molar mass fractions (>10^6^ g·mol^−1^) is used, resulting in higher process stability by balancing the decomposition process. The shortcoming of PHAs such as post-crystallization, which was still present in low molar mass grades with narrow distribution, was additionally overcome by the higher molar mass, and the benefits of low molar mass fractions for melt flowability were maintained. In addition, degradation was found to be rather counterproductive at narrow molar mass distributions, resulting in an unstable process and reducing process variability concerning productivity and variability of fiber diameter formation.

Indeed, the claimed benefits of PHBV towards PHB, such as lowering melt temperature, higher elasticity, or lower crystallinity, could not be confirmed or exploited for the meltblown process.

The results open up interesting approaches in the field of using PHB for meltblown applications. For future work, the aim is to generate PHB and PHBV with targeted broad molar mass distributions by combining the distributions of different grades or by blending a virgin grade with material from the recycling loop(s). Based on this study, subsequent research will also address the combination/blending of PHB with PHBV and PHV to investigate meltblown processability as well as the use of different PHBVs (“V-amounts”), as this field was only touched upon by the investigation in this work.

## Figures and Tables

**Figure 1 materials-16-06525-f001:**
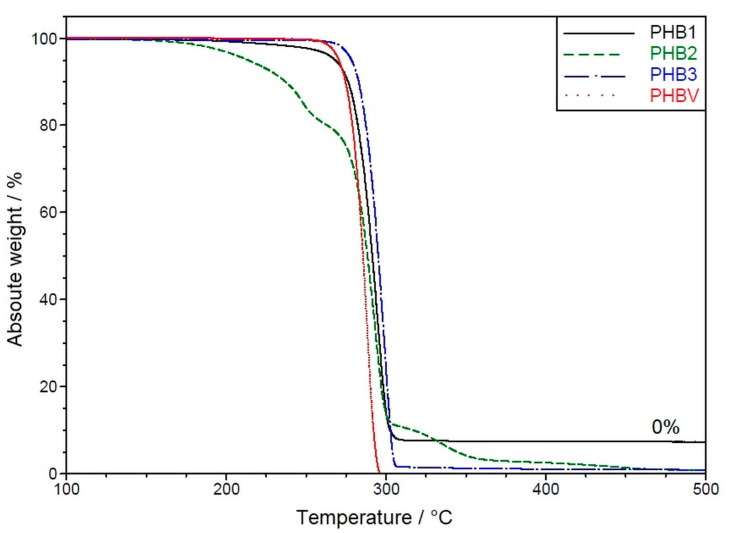
TGA heat ramp (10 K·min^−1^) of the different PHAs.

**Figure 2 materials-16-06525-f002:**
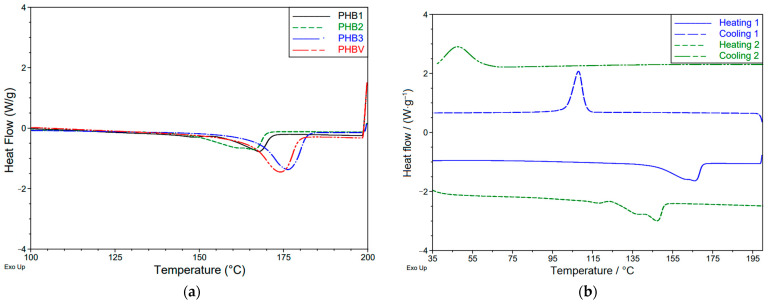
Representative excerpts of the DSC cycles of PHB (10 K·min^−1^ from RT to 200 °C); (**a**) 1st heating cycle of the different PHAs; (**b**) multiple heating/cooling runs of PHB2.

**Figure 3 materials-16-06525-f003:**
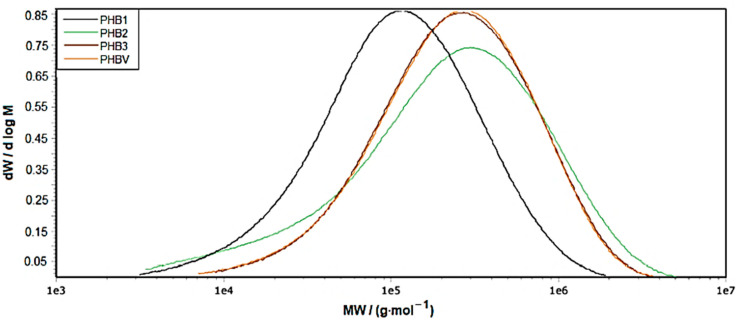
Molar mass distributions of the different PHA granules.

**Figure 4 materials-16-06525-f004:**
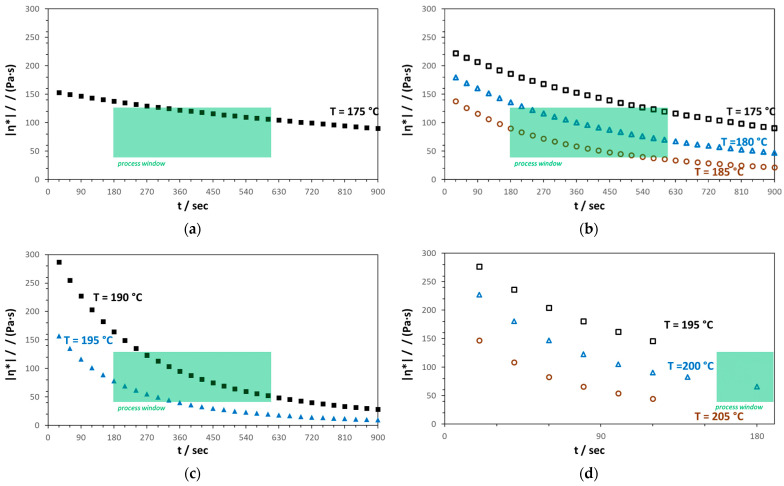
Absolute value of the complex shear viscosity (time sweep) of the different PHAs for different temperatures: (**a**) PHB1; (**b**) PHB2; (**c**) PHB3; (**d**) PHBV.

**Figure 5 materials-16-06525-f005:**
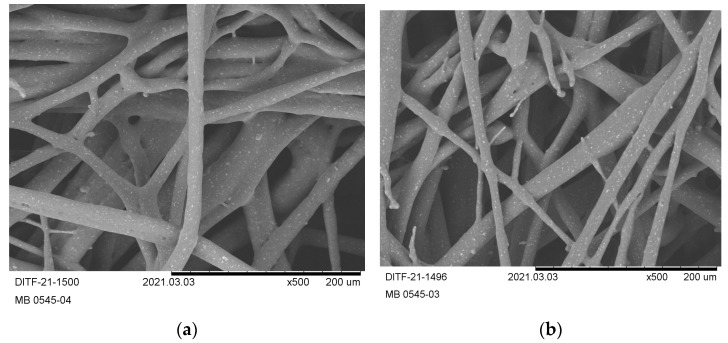
SEM images (magnification ×500) of the produced fabrics from PHB1: (**a**) PHB1-01, (**b**) PHB1-02, (**c**) PHB1-03, (**d**) PHB1-04.

**Figure 6 materials-16-06525-f006:**
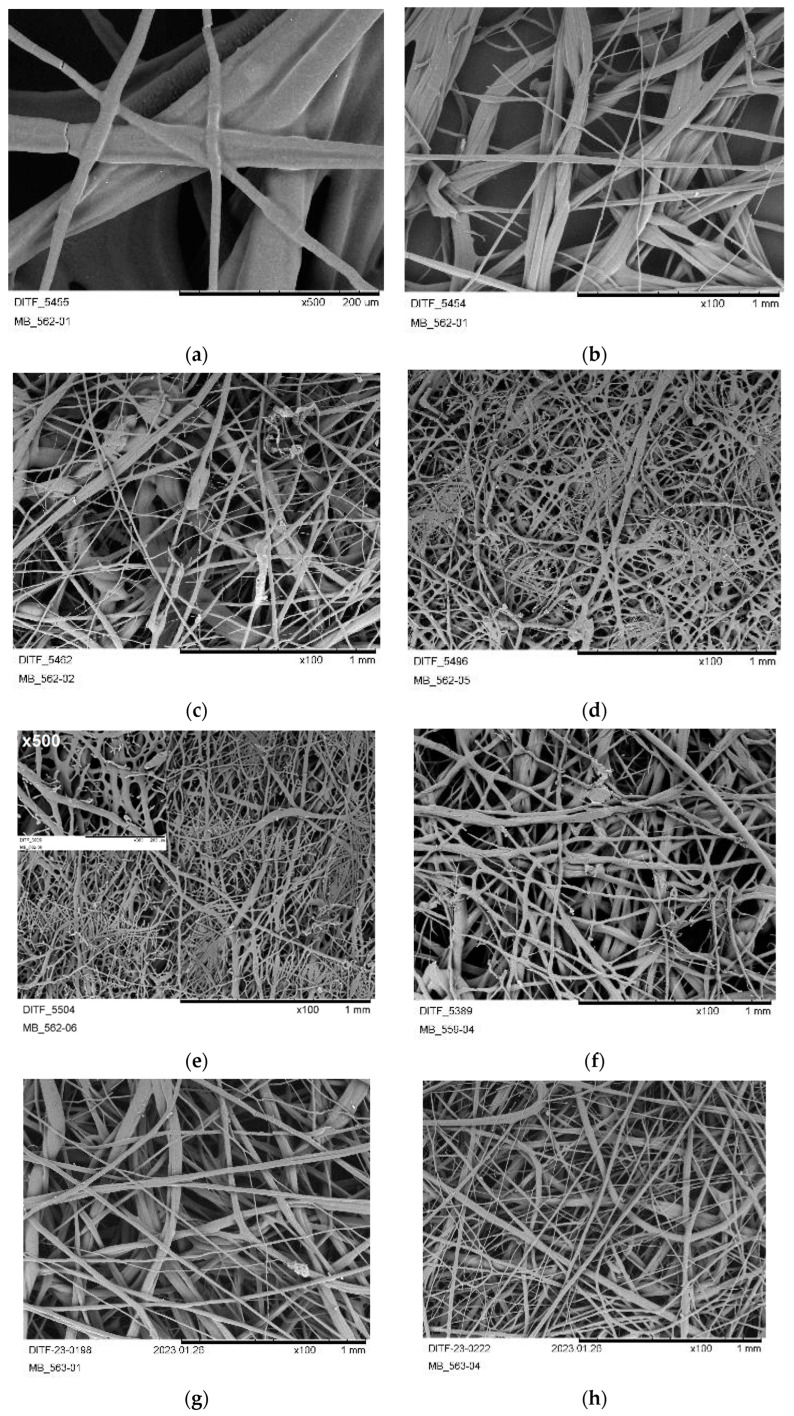
SEM pictures (×100) of the produced fabrics from PHB2: (**a**) PHB2-01; (**b**) PHB2-01; (**c**) PHB2-02; (**d**) PHB2-03; (**e**) PHB2-04 (×100 & ×500: small); (**f**) PHB2-05; (**g**) PHB2-06; (**h**) PHB2-07; (**i**) PHB2-08; (**j**) PHB2-09; (**k**) PHB2-10; (**l**) PHB2-11.

**Figure 7 materials-16-06525-f007:**
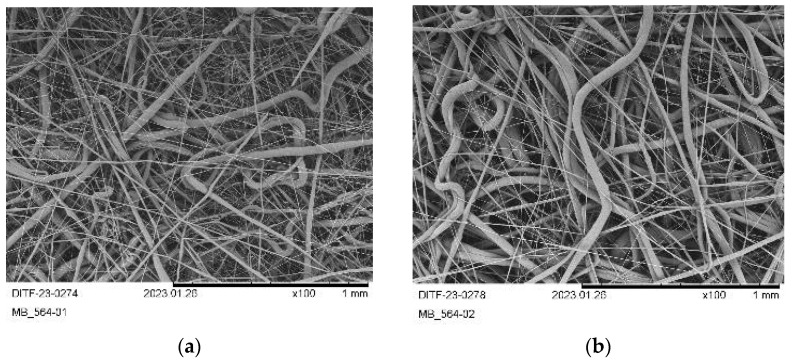
SEM pictures (×100) of the produced fabrics from PHB3: (**a**) PHB3-01; (**b**) PHB3-02; (**c**) PHB3-03; (**d**) PHB3-02 ×500.

**Figure 8 materials-16-06525-f008:**
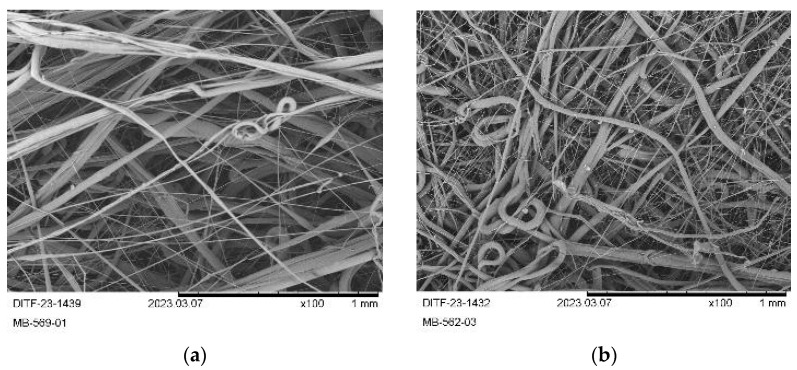
SEM pictures (×100) of the produced fabrics from PHBV: (**a**) PHBV-01; (**b**) PHBV-02.

**Figure 9 materials-16-06525-f009:**
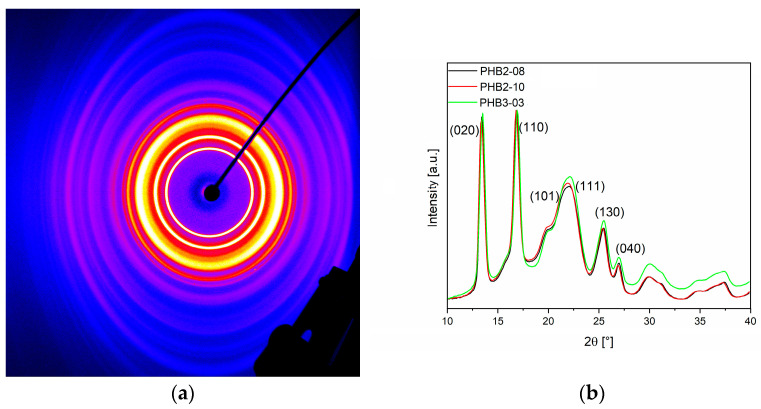
Exemplary X-ray scattering image (**a**) of a PHB meltblown sample and (**b**) corresponding X-ray diffraction patterns of different process settings; black: PHB2-08; red: PHB2-10; green: PHB3-03.

**Figure 10 materials-16-06525-f010:**
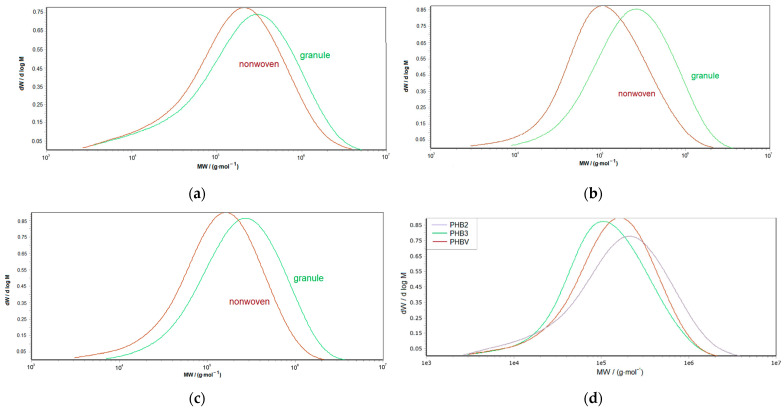
Comparison of the molar mass distributions of (**a**) granule and processed nonwoven of PHB2 (**b**) granule and processed nonwoven of PHB3 (**c**) granule and processed nonwoven of PHBV (**d**) nonwovens of PHB2, PHb3, and PHBV.

**Figure 11 materials-16-06525-f011:**
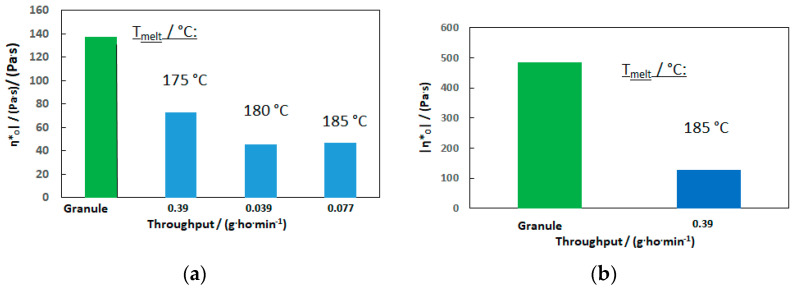
Comparison of complex shear viscosity of (**a**) granule and processed nonwoven of PHB2 (**b**) granule and processed nonwoven of PHB3.

**Table 1 materials-16-06525-t001:** Rheological characteristics from the time sweeps (ω = 10 rad·s^−1^, ε = 10%) and MFI measurements of the different PHAs.

Material:	PHB-1	PHB-2	PHB-3	PHBV
η_t0_ (175 °C)/(Pa·s):	153	222	697	12,700
η_to_ (180 °C)/(Pa·s):	129	137	484	434
G′_t0_ (180 °C)/Pa:	93	250	761	684
G″_t0_ (180 °C)/Pa:	1114	1354	4180	4287
MFI/(g·10 min^−1^) ^1^:	334	385	88	74

^1^ T = 190 °C, 2.16 kg.

**Table 2 materials-16-06525-t002:** Estimation of the process temperature and characteristic shear-rheological properties of the different PHAs at process temperature.

Material:	PHB-1	PHB-2	PHB-3	PHBV
T_proc_/°C:	175	(175) 180	(185) 190	(195) 200
η_to_ (T_proc)_/(Pa·s):	153	179	287	227
η_300__s_ (T_proc_)/(Pa·s):	127	110	113	49
G′_t__0_ (T_proc_)/Pa:	142	400	275	199
G″_t__0_ (T_proc_)/Pa:	1580	1753	2856	2262

**Table 3 materials-16-06525-t003:** Process settings for the meltblown of PHB1.

Trial	T_melt_/ °C	T_air_/°C	Throughput/(g·ho^−1^·min^−1^)	Die-Pressure/Bar	DCD/mm	Air Volume Flow/(Nm^3^·h^−1^)	Processable	Limitation(s)
PHB1-01	175	180	0.109	18.0	500	220	yes	Post-crystallization Fabrics not manageable due to crumbling disintegration
PHB1-02	175	180	0.077	12,6	500	220	yes	
PHB1-03	185	180	0.077	7.7	500	220	yes	
PHB1-04	185	180	0.109	12.9	500	220	Yes	
PHB2-01	175	180	0.077	25.4	350	325	yes	-
PHB2-02	175	180	0.077	25.0	350	220	yes	fibers flow together
PHB2-03	175	180	0.039	6.0	350	220	yes	fibers flow together
PHB2-04	175	180	0.039	6.0	350	325	limited	fabric glues to belt
PHB2-05	180	185	0.077	20.8	500	220	yes	shot formation
PHB2-06	180	175	0.077	21.7	500	220	Yes	-
PHB2-07	180	175	0.077	21.8	500	325	Yes	-
PHB2-08	180	175	0.039	8.4	500	220	Yes	-
PHB2-09	180	170	0.077	31.0	500	220	Yes	-
PHB2-10	180	165	0.077	27.4	500	220	Yes	-
PHB2-11	185	180	0.077	22.5	500	220	limited	Shot formation and gluing to conveyor
PHB3-01	185	180	0.039	35.0	500	325	yes	-
PHB3-02	185	180	0.039	38.7	500	220	limited	Adhesions at the die/shots
PHB3-03	190	185	0.051	36.6	500	325	yes	shots
PHBV-01	195	190	0.051	41.6	500	325	yes	-
PHBV-02	200	195	0.051	26.4	500	325	yes	-

**Table 4 materials-16-06525-t004:** Characteristics of the produced nonwoven fabrics; bold: samples showing the best/superior properties.

Sample	Base Weight/		Thickness/	Fiber Diameter		Air Permeability	Tenacity	Elongation ^1^	Modulus
	(g·m^2^)	CV/ %	µm	Median/µm	Mean/µm	/(l·m^−2^·h^−1^)	MD/CD /(N·mm^2^)	MD/CD /%	MD/CD /(N·mm^2^)
PHB1-01	- ^2^	- ^2^	- ^2^	9.7	11.4	- ^2^	- ^2^	- ^2^	- ^2^
PHB1-02	- ^2^	- ^2^	- ^2^	11.2	12.3	- ^2^	- ^2^	- ^2^	- ^2^
PHB1-03	- ^2^	- ^2^	- ^2^	10.4	10.9	- ^2^	- ^2^	- ^2^	- ^2^
PHB1-04	- ^2^	- ^2^	- ^2^	13.2	12.0	- ^2^	- ^2^	- ^2^	- ^2^
PHB2-01	87	22	252 ± 57	5.2	7.7	1710 ± 300	1.3 ± 0.2/0.5 ± 0.1	3 ± 1/3 ± 1	85 ± 13/27± 6
PHB2-02	91	16	360 ± 79	7.1	12.3	5910 ± 570	0.6 ± 0.1/0.3 ± 0.1	2 ± 1/2 ± 0	35 ± 3/25 ± 6
PHB2-03	86	11	451 ± 73	11.2	16.0	7760 ± 960	0.5 ± 0.1/0.3 ± 0.1	2 ± 0/2 ± 0	30 ± 2/28 ± 4
PHB2-04	88	13	290 ± 42	7.0	9.7	2200 ± 330	1.2 ± 0.1/0.6 ± 0	3 ± 1/3 ± 0	74 ± 8/32 ± 4
PHB2-05	105	16	331 ± 31	5.0	8.5	2860 ± 370	0.4 ± 0.1/0.3 ± 0	2 ± 1/2 ± 3	28 ± 4/25 ± 2
PHB2-06	94	14	346 ± 43	13.7	16.2	4640 ± 880	0.7 ± 0.1/0.5 ± 0.1	3 ± 1/3 ± 1	39 ± 7/28 ± 4
PHB2-07	99	16	299 ± 40	7.3	12.7	1550 ± 430	1.4 ± 0/0.7 ± 0.2	3 ± 0/3 ± 1	76 ± 5/38 ± 7
PHB2-08	93	11	235 ± 19	**4.6**	**7.0**	**680 ± 100**	**1.6** ± 0.1/**0.9** ± 0.1	**4** ± 1/**5** ± 1	**92** ± 4/**47** ± 2
PHB2-09	98	17	349 ± 41	17.1	19.6	4660 ± 1090	0.6 ± 0.2/0.4 ± 0.1	3 ± 0/3 ± 1	26 ± 4/28 ± 2
PHB2-10	96	14	374 ± 60	14.4	20.5	5730 ± 1400	0.8 ± 0.2/0.4 ± 0.1	3 ±1/3 ± 1	40 ± 6/24 ± 4
PHB2-11	80	9	194 ± 24	9.4	13.4	1500 ± 70	1.2 ± 0.2/0.5 ± 0.1	2 ± 0/2 ± 1	71 ± 15/35 ± 7
PHB3-01	98	14	406 ± 32	**2.6**	**5.0**	1620 ± 180	1.0 ± 0/0.7 ± 0.1	1 ± 0/2 ± 1	**96 ± 7**/41 ± 5
PHB3-02	100	14	476 ± 80	4.7	10.1	2670 ± 20	0.9 ± 0.1/0.7 ± 0.1	1 ± 0/2 ± 0	82 ± 11/44 ± 1
PHB3-03	95	15	409 ± 28	4.0	7.5	2160 ± 350	0.8 ± 0.1/0.6 ± 0.2	2 ± 1/2 ± 1	67 ± 3/45 ± 23
PHBV-01	126	14	661 ± 114	3.4	5.6	2800 ± 110	0.5 ± 0.1/0.5 ± 0.1	1 ± 1/2 ± 1	40 ± 6/40 ± 2
PHBV-02	120	8	566 ± 79	3.5	4.8	2030 ± 340	0.8 ± 0.1/0.3 ± 0	1 ± 0/1 ± 1	61 ± 5/31 ± 5

^1^ Elongation at max. force. ^2^ No measurement possible due to brittleness of the samples.

**Table 5 materials-16-06525-t005:** SEC and MFI data of processed meltblown material of PHB2, PHb3, and PHBV vs. SEC data of their virgin granules.

Material:	PHB-2	PHB-3	PHBV
Virgin granule:			
*M_n_*/(g·mol^−1^):	78,300	113,300	131,800
*M_w_*/(g·mol^−1^):	666,600	499,500	489,700
*∅*/-:	8.51	4.41	4.78
MFI (190 °C, 2.16 kg):	385	88	74
Meltblown material:			
*M_n_*/(g·mol^−1^):	65,500	58,600	66,500
*M_w_*/(g·mol^−1^):	402,300	203,500	236,200
*∅*/-:	6.14	3.47	3.55
MFI (190 °C, 2.16 kg):	>600 ^1^	222	201
MFI (185 °C, 2.16 kg):	482	-	-

^1^ Not measurable (exceeding device limit).

## Data Availability

Data are available on request due to privacy restrictions. The data presented in this study are available on request from the corresponding author. The data are not publicly available due to running project issues.

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
