# Peer review of "Influence of Rheological and Morphological Characteristics of Polyhydroxybutyrate on Its Meltblown Process Behavior"

_materials, 2023, doi:10.3390/ma16196525_

Round 1

Reviewer 1 Report

The manuscript entitled "Influence of rheological and morphological characteristics of polyhydroxybutyrate on its meltblown process behavior" deals with rheological and thermal characterization of different PHB commercial samples. Then, the processability of the considered materials,  through a meltblown process was assessed.

In my view, the proposed research looks very interesting, since the paper can provide a very usefull tool to gain important insights on the industrial exploitation of biopolymers. 

However, in my view, the manuscript needs some revisions before recommending its publication.

In particular:

- the different degradation mechanism of PHBV sample as compared to the other ones should be explained, also referring to already published papers.

- I'm not convinced about the sentence: "As revealed by the TGA curves, the polymers show different decomposition kinetics. Following from this, the data after the 1st heating cycle (1st cooling cycle as well as the second heat/cooling run) had to be excluded from further analyzation as they were affected by decomposition of the polymer, which was present for all four materials.". If during the first heating ramp the polymers experienced decomposition, the material tested during the following thermal cycles is not longer the native sample. Why DSC measurements were carried out to 250 °C?

- The Authors should comment about the lowest value ogf cristallinity of PHB1 sample, which has the lowest molar mass (as demonstrated by SEC).

- Figure 7 is quite confusing. I suggest to the Authors to split the reported curves in 2 o more different Figures, in order to improve the readability of the plotted data. Furthermore, the Authors should justify the selection of different temperatures for the rheological tests. The viscosity strictly depends on temperature; therefore, the comparison between the viscosity of different samples recorder at different temperature can be measleading. Finally, in my view, time sweep measurements are not very useful for assessing the polymer processability.

- Usually, zero-shear viscosity indicates the "Newtonian" viscosity of a polymeric material, evaluated by the fitting of the viscosity curve as a function of shear rate (or of the complex viscosity as a function of frequency, if the Cox-Merz rule apply) through a proper model (as an exemple, Cross or Carreau). Differently, in the manuscript, zero-shear viscosity refers to the initial value of viscosity rrecorded at the beginning of the time sweep test. Again, this issue can be measleading for the potential readers of the paper.

Author Response

We want to thank review 1 for his comments, which were critical on the one hand, but also very constructive on the other hand. We think they helped to get the manuscript more sharpened! Find following, the point-by-point response to all single comments:

Comment:

The manuscript entitled "Influence of rheological and morphological characteristics of polyhydroxybutyrate on its meltblown process behavior" deals with rheological and thermal characterization of different PHB commercial samples. Then, the processability of the considered materials,  through a meltblown process was assessed.

In my view, the proposed research looks very interesting, since the paper can provide a very usefull tool to gain important insights on the industrial exploitation of biopolymers. 

However, in my view, the manuscript needs some revisions before recommending its publication.

Respond:

We thank the reviewer for his time and effort. In the following, we reply to the comments step by step and highlight all changes made in our manuscript.

Comment:

In particular:

- the different degradation mechanism of PHBV sample as compared to the other ones should be explained, also referring to already published papers.

Respond:

Thank you for the comment. We add the explanation at l. 351f

In literature (e.g.[1,2,3]) the thermal degradation of PHB is referred to be almost exclusively a non-radical random chain scission reaction (cis-elimination) involving a six-membered ring transition state [1, 2]. This results in a rapid decrease in its molecular weight. [3] Further, the thermal stability was found to depend on the size of the counterion at the PHB endgroup, which have the form of carboxylic acid salts with Na+, K+, and Bu4N+ counterions. Based on that, the degradation via intermolecular α-deprotonation by carboxylate is suggested to be the main PHB decomposition pathway at moderate temperatures (above 120 °C). [4]

PHBV (poly(3-hydroxybutyrate-co-3-hydroxyvalerate) is one possible way to overcome this temperature sensitivity of PHB [3]. HY (hydroxyvalerate )-units in the polymer chain of the resulting copolymer are reported to lower the melting point and thus, the processing temperature.[5] It is also expected that PHBV is less affine to thermal degradation [6-8] However, the degradation action proposed for PHB was supposed to be also valid for PHBV with low HV contents. [5]

The PHBV in our study shows no lower melting point compared to a PHB of the same supplier. Further the degradation in TGA proceeds almost equally with increased temperature. However, the decomposition takes place even, slightly faster. Additionally, of note is that the TGA curve of PHBV is a one-step curve as the PHB, which is a further indicator that the PHBV contains only a marginal HV content (compare [7]). Moreover, the process temperature is slightly increased.

[1] Aoyagi, Y.; Yamashita, K.; Doi, Y. Thermal degradation of poly[(R)-3-hydroxybutyrate], poly[ε-caprolactone], and poly[(S)-lactide]. Polym. Degrad. Stab. 2002, 76, 53–59

[2] Erceg, M.; Kovacic, T.; Klaric, I. Thermal degradation of poly(3-hydroxybutyrate) plasticized with acetyl tributyl citrate. Polym. Degrad. Stab. 2005, 90, 313–318

 [3] Arrieta, M. P., Samper, M.D., Aldas, M., Lopez, J. On the Use of PLA-PHB Blends for Sustainable Food Packaging Applications 2017 Materials 10, 1008-1033.

[4] Kawalec, M.; Adamus, G.; Kurcok, P.; Kowalczuk, M.; Foltran, I.; Focarete, M.L.; Scandola, M.Carboxylate-induced degradation of poly(3-hydroxybutyrate)s. Biomacromolecules 2007, 8, 1053–1058.

[5] P. Bordes et al. / Polymer Degradation and Stability 94 (2009) 789–796

[6] Carrasco F, Dionisi D, Martinelli A, Majone M. Thermal stability of poly-hydroxyalkanoates. J Appl Polym Sci 2006;100(3):2111–2

[7] Li S-D, Yu PH, Cheung MK. Thermogravimetric analysis of poly(3-hydrox-ybutyrate) and poly(3-hydroxybutyrate-co-3-hydroxyvalerate). J Appl Polym Sci 2001;80(12):2237–44.

[8] He J-D, Cheung MK, Yu PH, Chen G-Q. Thermal analyses of poly(3-hydrox-ybutyrate), poly(3-hydroxybutyrate-co-3-hydroxyvalerate), and poly(3-hydroxybutyrate-co-3-hydroxyhexanoate). J Appl Polym Sci 2001;82(1):90–8.

Comment:

- I'm not convinced about the sentence: "As revealed by the TGA curves, the polymers show different decomposition kinetics. Following from this, the data after the 1st heating cycle (1st cooling cycle as well as the second heat/cooling run) had to be excluded from further analyzation as they were affected by decomposition of the polymer, which was present for all four materials.". If during the first heating ramp the polymers experienced decomposition, the material tested during the following thermal cycles is not longer the native sample. Why DSC measurements were carried out to 250 °C?

Respond:

Thank you for the critical comment. We recognized that our research design or even more the way it is presented here is misleading the understanding. We completely agree that the material is not the same as the native sample after the fist thermal cycle due to thermal decomposition. Exactly that we wanted to demonstrate additionally to our DSC tests by comparing the first to the second cycle respectively and additionally by comparing a second run to 200 °C with a second run to 250 °C. To note we carried out the DSC tests at 200°C and not at 250 °C. The run to 250 °C was just our first test, were we recognized the issue and execute the test from there on only at 200°C. However, also this impacted the material too much to obtain useable cooling (or second heating) runs.

We apologize for this and also corrected the methods part, were we left the 250 °C by mistake (l.256). The full plots are given below to show that the tests were executed respectively.

Figure of the DSC runs over the whole temperature range included in word file

We understood that our manuscript loses of its concept at this point and misleads our intention, which was to explain, why the cooling run and the second run were not used for further comparisons. We reduced figure 5 (now figure 3, l.364) and the respective paragraph at l.369 as follows.

fsee figure in word file

Figure 3. Representative excerpt of a multiple DSC heating/cooling run (10 K.min-1) of PHB2; (RT to 200 °C).

The samples show a recrystallization peak in the 1st cooling cycle. The melting peak in the second heating cycle is shifted to lower temperature and also changes into a double-peak structure. Moreover, the recrystallization peak is shifted significantly to lower temperature from around 110 °C to around 50 °C transforming from a sharp to a broader peak area. This behavior is representative for all materials used and can be attributed to the low degradation start temperature (170 °C) of PHB, so that the melt remains almost one hour (per run) above the decomposition temperature during the heating and cooling to/from 200 °C under the test conditions with 10 K.min.1. As revealed by the TGA curves, the polymers show different affiance to decomposition. Following from this, the data after the 1st heating cycle (1st cooling cycle as well as the second heat/cooling run) were excluded from further analyzation as they are affected by decomposition of the polymer, which was present for all four materials.

Comment:

- The Authors should comment about the lowest value ogf cristallinity of PHB1 sample, which has the lowest molar mass (as demonstrated by SEC).

Respond: Thank you for the comment. We added a comment to this at the respective paragraph at l.402f.

“It is also worth mentioning that the sample with the significant lowest molar mass (PHB1) also shows the lowest degree of crystallinity obtained by DSC (Table 2). It can be concluded that (a) the majority of polymer chains (< 10*5 g.mol-1) is too short to contribute significantly to crystal formation or (b) very short fractions present in the distribution (< 104) even hinder crystallization by acting as “spacer” between longer chains.“

Comment:

- Figure 7 is quite confusing. I suggest to the Authors to split the reported curves in 2 o more different Figures, in order to improve the readability of the plotted data. Furthermore, the Authors should justify the selection of different temperatures for the rheological tests. The viscosity strictly depends on temperature; therefore, the comparison between the viscosity of different samples recorder at different temperature can be measleading. Finally, in my view, time sweep measurements are not very useful for assessing the polymer processability.

Respond:

Thank you again for the constructive comment! We agree to split the plot into 4 different subfigures: one for each sample. Further we executed time-sweeps at different temperatures as the complex viscosity again were subjected to the impact of thermal decomposition over time. Normally the absolute value of complex viscosity in a temperature sweep is taken to estimate the process window for spinning processes. The degradation issued a lowering of the viscosity additionally to the decrease of viscosity with temperature, which cannot be considered separately from each other.

This is already stated in the paragraph at l. 442ff introducing the rheology results: “Due to the decomposition behavior of PHAs, already just above the melting temperature, it was not possible to execute measurements of amplitude sweeps, frequency-sweeps or temperature sweeps, which were not affected by the decline of the viscosity and elastic and loss moduli over measurement time. Due to that, time sweep at constant shear rate and strain were executed at different temperatures and for comparison of values only the time-unaffected start-values taken”

Comment:

- Usually, zero-shear viscosity indicates the "Newtonian" viscosity of a polymeric material, evaluated by the fitting of the viscosity curve as a function of shear rate (or of the complex viscosity as a function of frequency, if the Cox-Merz rule apply) through a proper model (as an exemple, Cross or Carreau). Differently, in the manuscript, zero-shear viscosity refers to the initial value of viscosity rrecorded at the beginning of the time sweep test. Again, this issue can be measleading for the potential readers of the paper.

Respond:

Thank you again for re-asking our procedure critically. As explained in the comment before, the complex viscosity was affected by degradation processes over time so that the typical plot of complex shear viscosity vs. shear rate in order to fit the zero shear-viscosity was not applicable. The most stable process point is the complex viscosity from time sweeps at t→0, so that the time- or degradation-unaffected viscosity, defined as ηt0 was used instead of the Newtonian viscosity η0.

To reduce confusion for the readers and avoid misunderstanding/misinterpretation we revised the introducing paragraph accordingly at l.442ff.:

“Due to the decomposition behavior of PHAs, already just above the melting temperature, it was not applicable to use the complex viscosity measurements from amplitude sweeps, frequency-sweeps or temperature sweeps as they were not affected by the decline of the viscosity and elastic / loss moduli over measurement time.

Due to that, time sweep at constant shear rate and strain were executed at different temperatures For comparison of initial properties of the different samples only the time-unaffected start-values were taken.”

Further we avoided to use the term “zero shear-viscosity” in the following and revised accordingly at l. 775, l.777, l.779 and l.818.

We want to thank reviewer 1 again for his useful comments, which were critical on the one hand, but also very constructive on the other hand. We think they helped to get the manuscript more sharpened!

Sincerely yours,

i.A. Dr.-Ing. TIm Höhnemann

i.A. M.Sc. Ingo Windschiegl

Reviewer 2 Report

The authors described their systematic study on melt-blowing PHB nonwovens using four different types of PHAs. While the paper was very comprehensive, it is required to proofread carefully to fix typos, format mistakes, and grammar issues. Examples includes but not limited to:

Line 103 “PLA-sheet” should be PLA-sheath”

Line 132 “to” should be “too”, preferably use “excessive”

Line 149 and 232 size exclusion calorimetry should be size exclusion chromatography

Line 154 add a “.” after “behavior”

Line 161-167, text needs to be taken to either before or after the table not sandwiching between table and footnote. Caption should be above table. Text mentioned two different PHB-PHV-compounds, but they were not in the table or in the results part.

Line 181 “steal” to “steel”

Line 215 despite PP,  PP is not used in the experiment.

Line 227, melt volume flow rate, should that be MFI, be consistent.

Line 263-266, sentences were broken. Machine direction (MD) was not defined in text.

Line 274 and 279, should be Figure 2.

There are numerous issues with grammar and spelling. 

Author Response

We thank reviewer 2 for his detailed comments. We considered all comments, but also revised the manuscript illusively and corrected further wording and spelling errors. All changes are marked respectively in the document.

Comment:

The authors described their systematic study on melt-blowing PHB nonwovens using four different types of PHAs. While the paper was very comprehensive, it is required to proofread carefully to fix typos, format mistakes, and grammar issues. Examples includes but not limited to:

Line 103 “PLA-sheet” should be PLA-sheath”

Respond:

Thank you for the comment! We revised accordingly!

Comment:

Line 132 “to” should be “too”, preferably use “excessive”

Respond:

Thank you for the comment! We revised accordingly!

Comment:

Line 149 and 232 size exclusion calorimetry should be size exclusion chromatography

Respond:

Thank you for the comment! We revised accordingly!

Comment:

Line 154 add a “.” after “behavior”

Respond:

Thank you for the comment! We revised accordingly!

Comment:

Line 161-167, text needs to be taken to either before or after the table not sandwiching between table and footnote. Caption should be above table. Text mentioned two different PHB-PHV-compounds, but they were not in the table or in the results part.

Respond:

Thank you for the comment! We revised accordingly!

The table jumped from its original position and was set back.

The PHB-PHV compounds had to be removed from the study due to issues with data availability agreement by the supplier. However, these compounds didn’t bring further contribution to this article’s findings. Our project is running on and it’s planned to report about them in a following article, when the works are finished.

Comment:

Line 181 “steal” to “steel”

Respond:

Thank you for the comment! We revised accordingly!

Comment:

Line 215 despite PP,  PP is not used in the experiment.

Respond:

Thank you for the comment! We revised accordingly!

Comment:

Line 227, melt volume flow rate, should that be MFI, be consistent.

Respond:

Thank you for the comment! We revised accordingly!

Comment:

Line 263-266, sentences were broken. Machine direction (MD) was not defined in text.

Respond:

Thank you for the comment! The paragraph was deleted in order by the revision process.

Comment:

Line 274 and 279, should be Figure 2.

Respond:

Thank you for the comment! We revised accordingly! However, the figure was deleted in order to the revision process and comment to reduce the amount of figures.

Sincerely yours,

i.A. Dr.-Ing. Tim Höhnemann

M.Sc. Ingo Windschiegl

Reviewer 3 Report

The problem considered at work is very important taking into account the technical aspects of processing of PHB. The manuscript is interesting. However, it should be indicated, what scientific problem was solved in the paper.

There are several points that I would like to address:

- what is the purpose and the novelty of the work,

- the figure on the first page of the paper  is not of good quality

- why were DSC tests performed before TGA tests?

- there are some punctuation and linguistic errors, i.e.line 49: double “and”, line 381: double “in”

- the discussion of the obtained results in comparison with the studies of other authors should be strengthened

- bibliographic references should be supplemented with the latest literature. Currently, only 6 positions are after 2020.

The English is good.

Author Response

We want to thank review 3 for his comments, will helped to improve the quality of our manuscript! Find following, the point-by-point response to all single comments:

Comment:

The problem considered at work is very important taking into account the technical aspects of processing of PHB. The manuscript is interesting. However, it should be indicated, what scientific problem was solved in the paper.

There are several points that I would like to address:

- what is the purpose and the novelty of the work,

Response:

Thank you for the comment! The scientific problem was to overcome the known and variously published shortcoming of PHB in thermoplastic fiber formation processes, related to our research the melt blowing. This was solved by screening PHBs, which are market-available in reasonable amounts for industrial purpose (which is not the case for numerously published co-polymers others than PHBV) and to reveal their differences.

We revised the articles abstract as follows in order to display the scientific problem and the novelty of the found results (l.8ff):

“PHB (Polyhydroxybutyrate) is a promising biopolymer. However, processing PHB in pure form in thermoplastic processes is limited due to its rapid degradation, very low initial crystallization rate, strong post-crystallization and its low final stretchability. In this article we screened commercial PHBs on morphological characteristics, rheological properties and “performance” in the meltblown process in order to reveal process-relevant properties and to overcome the shortcoming of PHB in thermoplastic processes for fiber formation. Evaluation of degradation (extruded (meltlbown) material vs. granules) was performed via rheological and SEC-analysis. The study revealed large differences in the minimum melt temperature (175 up to 200 °C) and grade-dependent limitation of accessible throughput on a 500 mm plant. Average fiber diameter could be lowered from around 10 μm to 2.4 μm in median, which are the finest reported values in literature so far. It was found that the determination of the necessary process temperature can be predicted well from the complex shear-viscosity. Different to expectations, it became apparent that a broader initial molar mass distribution (>8) is suitable to overcome the state-of-the art limitations of PHAs in order to stabilize the fiber formation, increase the productivity and obtain better resistance towards thermal degradation in process. Accordingly, longer polymer chain fractions could be more affected by degradation than medium and short polymer chains in the distribution. Further, a low initial narrow distributed molar mass reasoned in too brittle fabrics.“

Comment:

- the figure on the first page of the paper  is not of good quality

Response:

Thanks for this annotation. The quality of the TOC picture was revised and is of better quality in the resubmission. However, we deleted the figure from the word file as it is not necessary to have the TOC in the document and it needs to be cropped in size to fit into the file.

Comment:

- why were DSC tests performed before TGA tests?

Response:

Thank you for the question. Indeed, we didn’t perform DSC tests before TGA tests. We presented the TGA results before the DSC results, but both tests were executed in parallel for all used polymers. However, by our understanding, it should make no difference which test was performed first, as long as the results of both tests flow into an comprehensive evaluation.

Comments:

- there are some punctuation and linguistic errors, i.e.line 49: double “and”, line 381: double “in”

Response:

Thanks for the comment. We corrected the errors. We also checked the manuscript for further errors, wording, grammar and spelling errors. All changes are marked respectively in the document.

Comment:

- the discussion of the obtained results in comparison with the studies of other authors should be strengthened

Response:

 Thank you for this comment. We strengthened the comparison to the studies of other authors in the article. All changes are marked respectively in the manuscript: revision of abstract (l.8), new paragraph at l 347 and revision/extension of the paragraph at l. 649ff and a t l.673

Comment:

- bibliographic references should be supplemented with the latest literature. Currently, only 6 positions are after 2020.

Response:

Thank you for your comment! We understand the intention of the comment, but in our purpose our literature research represents the state of the technique, published for PHB. Especially for the meltblown processing of PHB no further peer-reviewed literature than the cited could be found so far. We know about further/other research activities on this side and presentations at scientific conferences, e.g. by the research institute NIRI (e.g. Dr. Ross Ward at the EDANA Juväskülä, 05/2023) or the companies Mango materials (Figen Seli at Dornbirn GFC, 09/2023) and CJ Biomaterials (Hugo Vuurens at AMI agricultural film conference, 03/2023). However, there are no published works from them in peer-reviewed literature so far.

Sincerely yours,

i.A. Dr.-Ing. Tim Höhnemann

M.Sc. Ingo Windschiegl

Round 2

Reviewer 1 Report

I recommend the pubblication of the manuscript as it stands, as the Authors intensively modified it following the suggestions of the Reviewers.

Author Response

Thank you very much!